# ACTIVE DOMAIN ADAPTATION OF MEDICAL IMAGES USING FEATURE DISENTANGLEMENT

## ABSTRACT

State-of-the-art deep learning models often fail to generalize in the presence of distribution shifts between training (source) data and test (target) data. Domain adaptation techniques have been developed to address this challenge, leveraging either labeled data (supervised domain adaptation) or unlabeled data (unsupervised domain adaptation). The careful selection of target domain samples can significantly enhance model performance and robustness, while also reducing the overall data requirements. Active learning aims to maximize performance with fewer annotations. In this paper, we introduce an innovative method for active learning in the presence of domain shifts. We propose a novel feature disentanglement approach to decompose image features into domain-specific and task-specific components. Thereafter we define multiple novel cost functions that identify informative samples under domain shift. We test our proposed method for medical image classification using one histopathology dataset and two chest x-ray datasets. Experiments show our proposed approach achieves state-of-the-art performance when compared to both domain adaptation methods and other active domain adaptation techniques.

## 1 INTRODUCTION

Deep neural networks (DNNs) demonstrate state-of-the-art (SOTA) results for many medical image analysis applications. Although they excel at learning from large labeled datasets, it is challenging for DNNs to generalize the learned knowledge to new target domains (Saenko et al., 2010; Torralba & Efros, 2011). This limits their real-world utility, as it is impractical to collect large datasets for every novel application with the aim of retraining the network. It is impractical to annotate every data point and include them as part of the training set since annotating medical images requires high clinical expertise. Considering a supervised domain adaptation setting, all available samples from the target class are not equally informative, and annotating every sample may result in a waste of time and effort. As such, it makes sense to select the most informative target domain samples for labeling. Additionally, in an unsupervised domain adaptation setting where sample annotation is not required, informative sample selection identifies the most important samples that are used for model training leading to improved performance than conventional approaches.

Active Learning (AL) methods enable an expert to select informative samples and add them to a training set for incremental training. This allows a model to obtain high performance with minimal labeled samples (i.e., high learning rates) and is particularly suitable for medical image analysis tasks where AL methods must adapt to varying conditions like device vendor, imaging protocol, machine learning model, etc. While conventional AL methods (Ash et al., 2019; Ducoffe & Precioso, 2018; Sener & Savarese, 2018) have extensively studied the problem of identifying informative instances for labeling, they typically choose samples from the same domain and hence do not account for domain shift. Thus conventional AL models are not very effective for domain adaptation applications. In many practical scenarios, models are trained on a source domain and deployed in a different target domain. This can be a common occurrence for medical image analysis applications where target domain data is scarce, or a result of different image-capturing protocols, parameters, devices, scanner manufacturers, etc. Domain shift is also observed when images of the dataset are from multiple facilities. Consequently, some domain adaptation techniques have to be used for such scenarios (Ganin & Lempitsky, 2015; Hoffman et al., 2018; Saenko et al., 2010).

Despite the existence of domain adaptation methods, the primary challenge of obtaining labeled data is accessing the right experts. In this scenario, it is appropriate that we make optimal use of the experts' time and use AL to obtain maximal information from minimal annotations. Hence it is beneficial to have a technique that can choose informative samples despite the observed domain shift. This will lead to different trained models being adapted for a wide variety of tasks in supervised and unsupervised domain adaptation.

In this work, we study the problem of active learning under such a domain shift, called Active Domain Adaptation (Kirsch et al., 2019) (ADA). Given i) labeled data in a source domain, ii) unlabeled data in a target domain, and iii) the ability to obtain labels for a fixed budget of target instances, the goal of ADA is to select target instances for labeling and learn a model with high accuracy on the target test set. We specifically apply our method to medical imaging datasets since domain shift is a common problem for medical diagnosis tasks.

## 2 PRIOR WORK

**Domain Adaptation In Medical Image Analysis:** Domain adaptation (DA) has attracted increasing attention in machine learning based medical image analysis (Ghafoorian et al., 2017; Raghu et al., 2019; Kamnitsas et al., 2017b). Its practical use includes segmenting cardiac structures from MR and CT images (Zhuang & Shen, 2016), and stain normalization of histopathology images from different hospitals (Bandi et al., 2019). The survey paper of (Guan & Liu, 2021) categorizes DA methods under 6 types. However, since our work focuses on supervised, and unsupervised DA we briefly review related works and refer the reader to (Guan & Liu, 2021) for more details.

**Supervised Domain Adaptation:** One of the first SDA methods for medical images (Kumar et al., 2017) used ResNet as the feature extractor and applied to mammography images. Huang et al. (Huang et al., 2017) propose to use LeNet-5 to extract features of histological images from different domains for epithelium-stroma classification, project them onto a subspace(via PCA) and align them for adaptation. Ghafoorian et al. (Ghafoorian et al., 2017) evaluate the impact of fine- tuning strategies on brain lesion segmentation, by using CNN models pre-trained on brain MRI scans. Their experimental results reveal that using only a small number of target training examples for fine-tuning can improve the transferability of models.

**Unsupervised Domain Adaptation:** UDA for medical image analysis has gained significance in recent years since it does not require labeled target domain data. Prior works in UDA focused on medical image classification (Ahn et al., 2020), object localisation, lesion segmentation (Heimann et al., 2013; Kamnitsas et al., 2017a), and histopathology stain normalization (Chang et al., 2021). Heimann et al. (Heimann et al., 2013) used GANs to increase the size of training data and demonstrated improved localisation in X-ray fluoroscopy images. Likewise, Kamnitsas et al. (Kamnitsas et al., 2017a) used GANs for improved lesion segmentation in magnetic resonance imaging (MRI). Ahn et al. (Ahn et al., 2020) use a hierarchical unsupervised feature extractor to reduce reliance on annotated training data. Chang et al. (Chang et al., 2021) propose a novel stain mix-up for histopathology stain normalization and subsequent UDA for classification. Graph networks for UDA (Ma et al., 2019; Wu et al., 2020) have been used in medical imaging applications (Ahmedt-Aristizabal et al., 2021) such as brain surface segmentation (Gopinath et al., 2020) and brain image classification (Hong et al., 2019a;b). However, none explore DA with active learning.

**Deep Active Learning In Medical Image Analysis:** (Wang et al., 2017a) use sample entropy, and margin sampling to select informative samples while (Zhou et al., 2016) use GANs to synthesize samples close to the decision boundary and annotate it by human experts. (Mayer & Timofte, 2018) use GANs to generate high entropy samples which are used as a proxy to find the most similar samples from a pool of real annotated samples. (Yang et al., 2017) propose a two-step sample selection approach based on uncertainty estimation and maximum set coverage similarity metric. Test-time Monte-Carlo dropout (Gal et al., 2017) has been used to estimate sample uncertainty, and consequently select the most informative ones for label annotation (Gal et al., 2017; Bozorgtabar et al., 2019). The state-of-the-art in active learning is mostly dominated by methods relying on uncertainty estimations. However, the reliability of uncertainty estimations has been questioned for deep neural networks used in computer vision and medical imaging applications due to model

calibration issues (Abdar et al., 2021; Jungo et al., 2020). Recent work (Budd et al., 2021; Mahapatra et al., 2021) has highlighted the importance of interpretability for medical imaging scenarios.

**Active Domain Adaptation**    ADA can be cost-effective solution when the quantity or cost (e.g. medical diagnosis) of labeling in the target domain is prohibitive. Despite its practical utility, ADA is challenging and has seen limited exploration since its introduction (Chattopadhyay et al., 2013; Kirsch et al., 2019). (Kirsch et al., 2019) first applied ADA to sentiment classification from text data by sampling instances based on model uncertainty and a learned domain separator. (Chattopadhyay et al., 2013) select target instances and learn importance weights for source points through a convex optimization problem.

In a traditional AL setting informative sample selection does not focus on addressing domain shift. Thus, AL methods based on uncertainty or diversity sampling are less effective for ADA. Uncertainty sampling selects instances that are highly uncertain under the model's beliefs (Gal et al., 2017), which under a domain shift leads to miscalibration and selection of uninformative, outlier, or redundant samples for labeling (Ovadia et al., 2019). AL based on diversity sampling selects instances dissimilar to one another (Gissin & Shalev-Shwartz, 2019; Sener & Savarese, 2018; Sinha et al., 2019). In ADA this can lead to sampling uninformative instances from regions of the feature space that are already well-aligned across domains (Prabhu et al., 2021). Although using uncertainty or diversity sampling exclusively is suboptimal for ADA, their combination can be very effective as shown by AADA (Su et al., 2020a). (Ash et al., 2019) perform clustering in a hallucinated "gradient embedding" space, but rely on distance-based clustering in high-dimensional spaces, which often leads to suboptimal results. (Prabhu et al., 2021) propose a label acquisition strategy, termed as Clustering Uncertainty-weighted Embeddings (CLUE), for ADA that combines uncertainty and diversity sampling without the need for complex gradient or domain discriminator-based diversity measures.

**Our contributions:**    While domain adaptation and active learning have been well explored in medical image analysis, their combination has not seen much work. Our work is one of the first to look at active domain adaptation in the medical image analysis setting. This paper makes the following contributions: **1**) We propose one of the first applications of active domain adaptation in medical image analysis. **2**) We propose a feature disentanglement approach that extracts class specific and class agnostic features from a given image. **3**) Using the different feature components we propose novel metrics to quantify the informativeness of samples across different domains. **4**) We demonstrate that the novel feature disentanglement components are able to identify informative samples across different domains.

## 3    METHOD

We aim to show the relevance of ADA in supervised domain adaptation (SDA) and unsupervised domain adaptation (UDA). While active SDA (ASDA) requires selecting the target samples to be labeled, in active UDA (AUDA) we select the unlabeled target samples that will go into the pool for training along with labeled samples from the source domain. Both ASDA and AUDA are related tasks requiring the selection of informative samples, and hence can be solved within a common framework. Although the individual components of ADA - addressing domain shift and informative sample selection - have been explored in detail, their combination presents a different challenge. In prior work much effort has gone into exploring properties such as transferable diversity, uncertainty, etc, wherein the common criteria for informative sample selection is adapted to ensure it does equally well for samples from a new domain. In our approach we propose to use feature disentanglement to extract different types of features from the labeled source data and unlabeled target data such that samples with the same label are projected to the same region of the new feature space.

Prior work on feature disentanglement for domain adaptation decompose the latent feature vector into domain specific and domain agnostic features. These approaches are sub-optimal for ADA because : 1) the domain agnostic features are usually segmentation maps that encode the structural information. However selecting informative images onthis basis is challenging due to different appearance and field of views captured by the target domain images. 2) domain specific features usually encode information such as texture, intensity distributions, etc, which are not generally useful in selecting informative samples from a different domain.

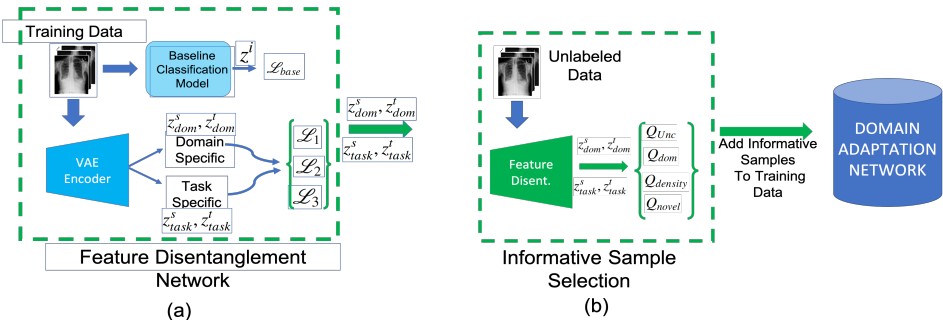

Figure 1: Workflow of the proposed method. (a) **Feature Disentanglement:** Training data goes through an autoencoder to obtain different components $z_{dom}, z_{task}$ and they are used to obtain different loss functions. After training is complete we get the disentangled features. (b) For **informative sample selection** we obtain disentangled feature representations of the unlabeled data and calculate the informativeness score of each sample in the batch. Thereafter the most informative samples are added to the labeled samples to initiate the domain adaptation steps.

Given source and target domains $S$ and $T$, an ideal domain independent feature's classification accuracy on domain $S$ is close to those obtained using features from a optimal classifier. At the same the features should be similar for samples having same labels from domains $S, T$. The resulting feature space can be used to train a classifier on domain $S$, and use conventional active learning techniques to select informative samples from domain $T$. In Active DA, the learning algorithm has access to labeled instances from the source domain $(X_S, Y_S)$, unlabeled instances from the target domain $X_{UT}$, and a budget $B$ which is much smaller than the amount of unlabeled target data. The learning algorithm may query an oracle to obtain labels for at most B instances from $X_{UT}$, and add them to the set of labeled target instances $X_{LT}$. The entire target domain data is $X_T = X_{LT} \bigcup X_{UT}$. The task is to learn a function $h : X \longrightarrow Y$ (a convolutional neural network (CNN) parameterized by $\theta$) that achieves good predictive performance on the target domain. The samples $x_S \in X_S$ and $x_T \in X_T$ are images, and labels $y_S \in Y_S, y_T \in Y_T$ are categorical variables $y \in 1, 2, \cdots, C$.

### 3.1 FEATURE DISENTANGLEMENT NETWORK:

Figure 1 shows the workflow of our proposed method. The feature disentanglement network (FDN) consists of an autoencoder each for source and targer domains. The FDN consists of two encoders $(E_S(\cdot), E_T(\cdot))$ and two decoder networks $(G_S(\cdot), G_T(\cdot))$, for the source and target domains respectively. Similar to a classic autoencoder, each encoder, $E_\bullet(\cdot)$, produces a latent code $z_i$ for image $x_i^\bullet \sim p_\bullet$. Each decoder, $G_\bullet(\cdot)$, reconstructs the original image from $z_i$. Furthermore, we divide the latent code, $z_i$, into two components: a domain specific component, $z_{dom}$, and a task specific component, $z_{task}$. The disentanglement network is trained using the following loss function:

$$\mathcal{L}_{Disent} = \mathcal{L}_{Rec} + \lambda_1 \mathcal{L}_1 + \lambda_2 \mathcal{L}_2 + \lambda_3 \mathcal{L}_3 + \lambda_{base} \mathcal{L}_{base} \tag{1}$$

$\mathcal{L}_{Rec}$, is the commonly used image reconstruction loss and is defined as:

$$\mathcal{L}_{Rec} = \mathbb{E}_{x_i^S \sim p_S} \left[ \left\| x_i^S - G_S(E_S(x_i^S)) \right\| \right] + \mathbb{E}_{x_j^T \sim p_T} \left[ \left\| x_j^T - G_T(E_T(x_j^T)) \right\| \right] \tag{2}$$

The disentangled features from both domains are denoted as $z_{dom}^s, z_{task}^s$ for source domain and $z_{dom}^t, z_{task}^t$ for the target domain. $z_{dom}^s, z_{task}^s$ are combined and input to the source decoder $G_S$ to get back the original source domain image, while $z_{dom}^t, z_{task}^t$ are combined and input the source decoder $G_T$ to get back the original target domain image. Since domain specific features encode information unique to the domain, they will be different for source and target domains. Hence the semantic similarity between $z_{dom}^t$ and $z_{dom}^s$ will be low. This is captured using

$$\mathcal{L}_1 = \langle z_{dom}^s, z_{dom}^t \rangle \tag{3}$$

where $\langle \cdot \rangle$ denotes the cosine similarity of the two feature vectors. Additionally, we expect that the task specific features of the two domains will have high similarity which is incorporated using:

$$\mathcal{L}_2 = 1 - \langle z_{task}^s, z_{task}^t \rangle \tag{4}$$

We want the two components of the features to be as dissimilar as possible in order to capture mutually complementary information, and this is achieved using the following loss term

$$\mathcal{L}_3 = \langle z_{dom}^s, z_{task}^s \rangle + \langle z_{dom}^t, z_{task}^t \rangle \tag{5}$$

Additionally, since the task specific features encode information specific to the task, they should be such that when using it to train a classifier it should give performance levels similar to those obtained using the original image features. Our approach is to ensure that the task specific features are such that they do not result in a performance drop. We first train a baseline classification model $M_{base}$ using the original image features of source domain data (since labels are available only for the source domain data). For a given sample $i$ from source domain, we pass it through $M_{base}$ to obtain a feature vector $z^i$. The task specific component is denoted as $z_{task}^s$. Their cosine similarity should be high to ensure it captures highly relevant information and the corresponding loss term is

$$\mathcal{L}_{base} = 1 - \langle z_{task}^s, z^i \rangle \tag{6}$$

Note that the feature vector $z^i$ is obtained from a pre-trained classifier and can be considered as the optimal feature vector (although that depends upon the chosen classifier model $M_{base}$) whereas the task specific feature vector $z_{task}^s$ is obtained as part of the training process. Our objective is to ensure that $z_{task}^s$ is very close to $z^i$ in terms of semantic similarity. $\mathcal{L}_{base}$ is named as such to denote its comparison with $M_{base}$. We use a DenseNet-121 as $M_{base}$.

## 3.2 Informative Sample Selection

We train two classifiers, $M_{task}^{source}$ on $z_{task}^s$, and $M_{dom}^{source}$ on $z_{dom}^s$. By design $z_{task}^s$ has high similarity $z_{task}^t$ which ensures that $M_{task}^{source}$ trained with $z_{task}^s$ can be used with $z_{task}^t$ to obtain similar performance levels, and also identify informative samples. We use the following criteria to choose informative samples: 1) **Uncertainty:** - We take $M_{task}^{source}$ trained on $z_{task}^s$ and use it to calculate the uncertainty of the target domain samples with $z_{task}^t$. To measure informativeness we use predictive entropy $\mathcal{H}(Y|x)$ (Wang & Shang, 2014) which for C-way classification, is defined as:

$$Q_{Unc} = \mathcal{H}(Y|x) = -\sum_{c=1}^{C} p_\theta(Y = c|x) \log p_\theta(Y = c|x) \tag{7}$$

2) **Domainness:** determines whether a sample is from the same domain as a reference sample (e.g., source domain). Recent active learning or ADA methods (Huang et al., 2018; Su et al., 2020b) consider samples with higher distinctiveness from source domain samples as informative since they capture the unique characteristics in the target domain. This approach is susceptible to choosing outliers. For a given target domain sample we obtain $z_{task}^t, z_{dom}^t$. $z_{dom}^t$ is compared with $z_{dom}^s$ of each label. If the cosine similarity is below a threshold then the sample is determined as different from the source domain data, and hence not considered for labeling. Fu et al (Fu et al., 2021) show that too low similarity scores between source and target domain samples denotes outliers and too high similarity scores indicate uninformative samples since the target domain sample has already been included within the training set. Thus we define a score

$$Q_{dom} = \begin{cases} 0 & \text{if } \langle z_{dom}^s, z_{dom}^t \rangle < \eta_1 \\ \langle z_{dom}^s, z_{dom}^t \rangle & \text{if } \eta_1 \leq \langle z_{dom}^s, z_{dom}^t \rangle \leq \eta_2 \\ 0 & \text{if } \langle z_{dom}^s, z_{dom}^t \rangle > \eta_2 \end{cases} \tag{8}$$

To set the thresholds $\eta_1$ and $\eta_2$ we plot a distribution of the cosine similarity values and $\eta_1$ equals the 30th percentile value while $\eta_2$ corresponds to the 75th percentile.

3) **Density:** - determines whether a sample represents other samples which are similar in the feature space. The number of annotations can be reduced by labeling samples which are representative of many other samples. If a sample lies in a dense region of the feature space then it is representative of many other samples. We cluster the target domain samples into $N$ clusters using the task specific features $z_{task}^t$ where $N$ is the number of classes of the source domain data. For each sample we calculate the feature distance with respect to other samples in the batch, and take the average distance over the top $K$ closest samples. A higher average feature distance indicates that the sample is more

similar to other samples and is in a dense region of the feature space. We set $K = 20$. We define a density score as this average feature distance:

$$Q_{density} = \frac{1}{K} \sum_{k=1,\cdots,K} \langle z_{task}^i, z_{task}^k \rangle \tag{9}$$

4) **Novelty:**- This criterion checks whether the selected target sample for labeling is similar or different to previously labeled target samples. For a given target domain sample $i$ with feature vector $z_{task}^i$ we calculate its cosine similarity with previously annotated samples $z_{task}^j$. If the similarity is high then the informativeness score of sample $i$ is low and vice-versa. Thus we define a "novelty-score":

$$Q_{novel} = \sum_j 1 - \langle z_{task}^i, z_{task}^j \rangle \tag{10}$$

The cosine similarities of $i$ with other previously annotated samples $j$ are summed to get the "novelty-score". The final informativeness score of a sample is the sum of all the above scores:

$$Q_{Inf} = \lambda_{Unc} Q_{Unc} + \lambda_{Dom} Q_{Dom} + \lambda_{Density} Q_{Density} + \lambda_{Novel} Q_{Novel} \tag{11}$$

Higher values of $Q_{Inf}$ indicates greater informativeness. The top $N$ informative samples are selected in every batch and added to the training set, and the classifier is updated. Informative sample selection continues till there is no further change in the performance of a separate validation set.

## 4 EXPERIMENTAL RESULTS

**Baseline Methods:** We compare our proposed method against several state-of-the art methods for Active DA and Active Learning such as: 1) AADA: Active Adversarial Domain Adaptation (AADA) (Su et al., 2020a); 2) Entropy (Wang & Shang, 2014): Selects instances for which the model has highest predictive entropy; 3) BADGE (Ash et al., 2019): a state-of-the-art AL using KMeans++ on "gradient embeddings" to incorporate model uncertainty and diversity. 4) CLUE method of (Prabhu et al., 2021) - Clustering Uncertainty-weighted Embeddings (CLUE) using uncertainty-weighted clustering. under the model and diverse in feature space. 5) (Ma et al., 2021)'s active universal domain adaptation. 6) (Fu et al., 2021)'s method using transferable uncertainty.

**Ablation Studies:** We also show the results for ablation with the following methods: 1) $\text{AUDA}_{w/\mathcal{L}_1}$: Our proposed method used for AUDA without the loss term $\mathcal{L}_1$ in Eqn.3; 2) $\text{AUDA}_{w/\mathcal{L}_2}$: AUDA without the loss term $\mathcal{L}_2$ in Eqn.4; 3) $\text{AUDA}_{w/\mathcal{L}_3}$: AUDA without the loss term $\mathcal{L}_3$ in Eqn.5; 4) $\text{AUDA}_{w/\mathcal{L}_{base}}$: AUDA without the loss term $\mathcal{L}_{base}$ in Eqn.6; 5) $\text{AUDA}_{w/Q_{Unc}}$: AUDA without the informativeness term $\text{Q}_{Unc}$ in Eqn.7; 6) $\text{AUDA}_{w/Q_{dom}}$: AUDA without the domainness term $\text{Q}_{dom}$ in Eqn.8; 7) $\text{AUDA}_{w/Q_{density}}$: AUDA without the density term $\text{Q}_{density}$ in Eqn.9; 8) $\text{AUDA}_{w/Q_{novel}}$: AUDA without the novelty term $\text{Q}_{novl}$ in Eqn.10;

### 4.1 EXPERIMENTAL SETTINGS

We use the source domain and part of the target dataset to train our feature disentanglement method, and use it on the remaining target domain samples to obtain $z_{task}^t$ and $z_{dom}^t$. Selected informative samples from the target domain are added to the training set which initially consists of only source domain samples. After each addition the classifier is updated and evaluated on a separate test set from the target domain. In the unsupervised setting there are no samples to label. We adapt our active domain adaptation method such that instead of using the entire unlabeled dataset, we select informative samples from target domain to use for training the classifier.

Since our goal is to demonstrate the effectiveness of our active learning method under domain shift and not to propose a new domain adaptation method, we show results of our method integrated with existing SOTA methods for SDA and UDA. We adopt the following experimental setup: **1)** Train a benchmark method in a fully-supervised manner with training, validation and test data from the same hospital/dataset. This setting gives the upper-bound performance expectation for a SDA model, which depends upon the network architecture. We refer to this benchmark as $FSL - SameDomain$

(fully supervised learning based method on same domain data). **2**) Train a SOTA domain adaptation method (either supervised DA or unsupervised DA) using the available source and target domain data, and without informative sample selection - $SDA_{SOTA}$ (SOTA supervised domain adaptation) or $UDA_{SOTA}$ (SOTA unsupervised domain adaptation). We observe that $SDA_{SOTA}$ is obtained by taking the $FSL - SameDomain$ and fine tuning the network using labeled target domain samples. **3**) We use our active domain adaptation (ADA) method to select informative samples and incrementally add to the training set. We investigate its effectiveness in selecting the best possible set of samples for labeling, and explore their degree of success in reducing the required number of annotated samples. As we add samples to the training set we report the test accuracy for every $10\%$ increase of the training set. **4**) We also report the performance of other active learning methods- including domain adaption based and conventional methods that do not address the domain shift.

## 4.2 RESULTS ON HISTOPATHOLOGY DATASETS

**Dataset Description:** We use the CAMELYON17 dataset (Bandi et al., 2019) to evaluate the performance of the proposed method on tumor/normal classification. In this dataset, a total of 500 $H\&E$ stained WSIs are collected from five medical centers (denoted as $C1, C2, C3, C4, C5$ respectively). 50 of these WSIs include lesion-level annotations. All positive and negative WSIs are randomly split into training/validation/test sets and provided by the organizers in a $50/30/20\%$ split for the individual medical centers to obtain the following split: $C1$:37/22/15, $C2$: 34/20/14, $C3$: 43/24/18, $C4$: 35/20/15, $C5$: 36/20/15. $256 \times 256$ image patches are extracted from the annotated tumors for positive patches and from tissue regions of WSIs without tumors for negative patches. We use $\lambda_1 = 0.8, \lambda_2 = 1.05, \lambda_3 = 0.85, \lambda_{base} = 0.95, \lambda_{Unc} = 0.9, \lambda_{Dom} = 0.75, \lambda_{Density} = 1.0, \lambda_{Novel} = 1.0, \eta_1 = 0.21, \eta_2 = 0.82$.

Since the images have been taken from different medical centers their appearance varies despite sharing the same disease labels. This is due to slightly different protocols of $H\&E$ staining. Stain normalization has been a widely explored topic which aims to standardize the appearance of images across all centers, which is equivalent to domain adaptation. Recent approaches to stain normalization/domain adaptation favour use of GANs and other deep learning methods. We compare our approach to recent approaches and also with (Chang et al., 2021) which explicitly performs UDA using MixUp. The method by (Chang et al., 2021) is denoted as $UDA_{SOTA}$

To evaluate our method's performance: We use $C1$ as the source dataset and train a ResNet-101 classifier (He et al., 2016) (ResNet$_{C1}$). Each remaining dataset from the other centers are, separately, taken as the target dataset. We select informative samples add them to training set and update ResNet$_{C1}$. As a baseline, we perform the experiment without domain adaptation denoted as $No - ADA$ where ResNet$_{C1}$ is used to classify images from other centers. All the above experiments are repeated using each of $C2, C3, C4, C5$ as the source dataset. We report in Table 1 a center wise and also an average performance performance for different UDA methods. The results in Table 1 show that UDA methods are better than conventional stain normalization approaches as evidenced by the superior performance of (Chang et al., 2021). In Table 2 we report performance of different active domain adaptation methods. The numbers are compared against the average for all 5 centers.

|  | No ADA | MMD | CycleGAN | Chang ($UDA_{SOTA}$) | FSL-Same Domain | $SDA_{SOTA}$ |
|---|---|---|---|---|---|---|
| $C1$ | 0.8068 | 0.8742 | 0.9010 | 0.964 | 0.976 | 0.969 |
| $C2$ | 0.7203 | 0.6926 | 0.7173 | 0.933 | 0.957 | 0.941 |
| $C3$ | 0.7027 | 0.8711 | 0.8914 | 0.931 | 0.95 | 0.938 |
| $C4$ | 0.8289 | 0.8578 | 0.8811 | 0.95 | 0.971 | 0.957 |
| $C5$ | 0.8203 | 0.7854 | 0.8102 | 0.927 | 0.942 | 0.933 |
| $Avg.$ | 0.7758 | 0.8162 | 0.8402 | 0.941 | 0.959 | 0.948 |

Table 1: Classification results in terms of AUC measures for different domain adaptation methods on CAMELYON17 dataset. $FSL - SD$ is a fully-supervised model trained on target domain data.

## 4.3 RESULTS ON CHEST XRAY DATASET

**Dataset Description:** We use the following chest Xray datasets: **NIH Chest Xray** Dataset: The NIH ChestXray14 dataset (Wang et al., 2017b) has $112,120$ expert-annotated frontal-view X-rays

| | 10% | 20% | 30% | 40% | 50% | 60% | 70% | 80% | 90% | 100% | p- |
|---|---|---|---|---|---|---|---|---|---|---|---|
| FSL-SD | 0.959 | 0.959 | 0.959 | 0.959 | 0.959 | 0.959 | 0.959 | 0.959 | 0.959 | 0.959 | $< 0.001$ |
| Random | 0.693 | 0.71 | 0.75 | 0.794 | 0.821 | 0.858 | 0.891 | 0.914 | 0.928 | 0.938 | $< 0.001$ |
| Unc | 0.706 | 0.733 | 0.772 | 0.812 | 0.845 | 0.891 | 0.922 | 0.931 | 0.939 | 0.943 | $< 0.001$ |
| AADA | 0.712 | 0.742 | 0.791 | 0.841 | 0.872 | 0.903 | 0.924 | 0.939 | 0.945 | 0.948 | 0.001 |
| BADGE | 0.707 | 0.728 | 0.768 | 0.803 | 0.847 | 0.885 | 0.903 | 0.924 | 0.932 | 0.940 | 0.005 |
| CLUE | 0.715 | 0.746 | 0.786 | 0.839 | 0.878 | 0.911 | 0.921 | 0.934 | 0.941 | 0.947 | 0.007 |
| Fu | 0.714 | 0.739 | 0.775 | 0.813 | 0.849 | 0.883 | 0.914 | 0.925 | 0.935 | 0.944 | 0.001 |
| Su | 0.721 | 0.754 | 0.793 | 0.825 | 0.858 | 0.889 | 0.914 | 0.929 | 0.941 | 0.95 | 0.02 |
| $Our_{ASDA}$ | 0.73 | 0.775 | 0.801 | 0.831 | 0.872 | 0.895 | 0.927 | 0.937 | 0.946 | 0.964 | 0.04 |
| $Our_{AUDA}$ | 0.721 | 0.762 | 0.793 | 0.828 | 0.863 | 0.893 | 0.915 | 0.927 | 0.941 | 0.952 | - |
| **Ablation Studies** | | | | | | | | | | | |
| **Feature Disentanglement** | | | | | | | | | | | |
| $AUDA_{w/\mathcal{L}_1}$ | 0.702 | 0.734 | 0.772 | 0.842 | 0.872 | 0.885 | 0.896 | 0.902 | 0.911 | 0.921 | 0.001 |
| $AUDA_{w/\mathcal{L}_2}$ | 0.711 | 0.729 | 0.765 | 0.802 | 0.854 | 0.867 | 0.881 | 0.898 | 0.914 | 0.928 | 0.005 |
| $AUDA_{w/\mathcal{L}_3}$ | 0.692 | 0.724 | 0.768 | 0.813 | 0.843 | 0.869 | 0.884 | 0.896 | 0.901 | 0.914 | 0.0009 |
| $AUDA_{w/\mathcal{L}_{base}}$ | 0.671 | 0.703 | 0.734 | 0.771 | 0.81 | 0.848 | 0.866 | 0.881 | 0.895 | 0.908 | 0.0008 |
| **Informative Sample Selection** | | | | | | | | | | | |
| $AUDA_{w/Q_{Unc}}$ | 0.705 | 0.74 | 0.778 | 0.852 | 0.881 | 0.898 | 0.906 | 0.913 | 0.924 | 0.932 | 0.001 |
| $AUDA_{w/Q_{dom}}$ | 0.691 | 0.724 | 0.761 | 0.812 | 0.857 | 0.884 | 0.898 | 0.904 | 0.916 | 0.923 | 0.001 |
| $AUDA_{w/Q_{density}}$ | 0.693 | 0.719 | 0.753 | 0.788 | 0.814 | 0.861 | 0.878 | 0.896 | 0.908 | 0.919 | 0.0001 |
| $AUDA_{w/Q_{novel}}$ | 0.682 | 0.711 | 0.746 | 0.779 | 0.817 | 0.856 | 0.869 | 0.882 | 0.897 | 0.912 | 0.0001 |

Table 2: **Active Domain Adaptation Results For Camelyon17 dataset**. AUC values for different baselines and proposed approach along with ablation studies.

from $30,805$ unique patients and has 14 disease labels. Original images were resized to $256 \times 256$, and we use $\lambda_1 = 0.85, \lambda_2 = 1.1, \lambda_3 = 0.95, \lambda_{base} = 1.2, \lambda_{Unc} = 1.1, \lambda_{Dom} = 0.9, \lambda_{Density} = 1.05, \lambda_{Novel} = 1.25, \eta_1 = 0.24, \eta_2 = 0.78$. **CheXpert** Dataset: This datset (Irvin et al., 2019) has $224,316$ chest radiographs of $65,240$ patients labeled for the presence of 14 common chest conditions. Original images were resized to $256 \times 256$, and we use $\lambda_1 = 0.95, \lambda_2 = 1.0, \lambda_3 = 1.1, \lambda_{base} = 1.0, \lambda_{Unc} = 1.2, \lambda_{Dom} = 1.1, \lambda_{Density} = 0.95, \lambda_{Novel} = 1.0, \eta_1 = 0.29, \eta_2 = 0.8$. These two datasets have the same set of disease labels.

We divide both datasets into train/validation/test splits on the patient level at $70/10/20$ ratio, such that images from one patient are in only one of the splits. Then we train a DenseNet-121 (Rajpurkar et al., 2017) classifier on one dataset (say NIH's train split). Here the NIH dataset serves as the source data and CheXpert is the target dataset. We then apply the trained model on the training split of the NIH dataset and tested on the test split of the same domain the results are denoted as $FSL - Same$. When we apply this model to the test split of the CheXpert data without domain adaptation the results are reported under No-$UDA$.

Table 3 show classification results for different DA techniques on NIH dataset as source domain and CheXpert as target domain. The reverse scenario results are shown in the Supplementary. UDA methods perform worse than $FSL - Same$ since it is very challenging to perfectly account for domain shift. However all UDA methods perform better than fully supervised methods trained on one domain and applied on another without domain adaptation. The DANN architecture (Ganin et al., 2016) outperforms MMD and cycleGANs, and is on par with graph convolutional methods GCAN (Ma et al., 2019) and GCN2 (Hong et al., 2019b). However our method outperforms all compared methods due to the combination of domain adaptation and informative sample selection.

| | Atel. | Card. | Eff. | Infil. | Mass | Nodule | Pneu. | Pneumot. | Consol. | Edema | Emphy. | Fibr. | PT. | Hernia |
|---|---|---|---|---|---|---|---|---|---|---|---|---|---|---|
| No DA | 0.697 | 0.814 | 0.761 | 0.652 | 0.739 | 0.694 | 0.703 | 0.781 | 0.704 | 0.792 | 0.815 | 0.719 | 0.728 | 0.811 |
| MMD | 0.741 | 0.851 | 0.801 | 0.699 | 0.785 | 0.738 | 0.748 | 0.807 | 0.724 | 0.816 | 0.831 | 0.745 | 0.754 | 0.846 |
| CycleGANs | 0.765 | 0.874 | 0.824 | 0.736 | 0.817 | 0.758 | 0.769 | 0.832 | 0.742 | 0.838 | 0.865 | 0.762 | 0.773 | 0.864 |
| DANN | 0.792 | 0.902 | 0.851 | 0.761 | 0.849 | 0.791 | 0.802 | 0.869 | 0.783 | 0.862 | 0.894 | 0.797 | 0.804 | 0.892 |
| FSL$-SD$ | 0.849 | 0.954 | 0.903 | 0.814 | 0.907 | 0.825 | 0.844 | 0.928 | 0.835 | 0.928 | 0.951 | 0.847 | 0.842 | 0.941 |
| $SDA_{SOTA}$ | 0.854 | 0.965 | 0.914 | 0.824 | 0.918 | 0.835 | 0.856 | 0.937 | 0.845 | 0.936 | 0.963 | 0.861 | 0.852 | 0.952 |
| GCN2 (UDA$_{SOTA}$) | 0.809 | 0.919 | 0.870 | 0.765 | 0.871 | 0.807 | 0.810 | 0.882 | 0.792 | 0.883 | 0.921 | 0.817 | 0.812 | 0.914 |

Table 3: Classification results on the CheXpert dataset's test split using NIH data as the source domain. Note: $FSL - SD$ is a fully-supervised model trained on target domain data.

| | 10% | 20% | 30% | 40% | 50% | 60% | 70% | 80% | 90% | 100% | p- |
|---|---|---|---|---|---|---|---|---|---|---|---|
| FSL-SD | 0.814 | 0.814 | 0.814 | 0.814 | 0.814 | 0.814 | 0.814 | 0.814 | 0.814 | 0.814 | < 0.001 |
| Random | 0.639 | 0.671 | 0.709 | 0.741 | 0.775 | 0.784 | 0.797 | 0.810 | 0.818 | 0.821 | < 0.001 |
| Unc | 0.648 | 0.687 | 0.725 | 0.763 | 0.797 | 0.809 | 0.819 | 0.835 | 0.842 | 0.851 | < 0.001 |
| AADA | 0.655 | 0.694 | 0.735 | 0.773 | 0.808 | 0.829 | 0.845 | 0.858 | 0.876 | 0.88 | < 0.001 |
| BADGE | 0.643 | 0.678 | 0.716 | 0.757 | 0.789 | 0.81 | 0.824 | 0.843 | 0.849 | 0.858 | 0.005 |
| CLUE | 0.648 | 0.688 | 0.729 | 0.763 | 0.793 | 0.815 | 0.837 | 0.849 | 0.863 | 0.869 | 0.007 |
| Fu | 0.652 | 0.689 | 0.732 | 0.775 | 0.805 | 0.827 | 0.845 | 0.855 | 0.872 | 0.879 | 0.001 |
| Su | 0.656 | 0.688 | 0.732 | 0.779 | 0.810 | 0.823 | 0.843 | 0.861 | 0.869 | 0.881 | 0.02 |
| $\text{Our}_{ASDA}$ | 0.669 | 0.702 | 0.743 | 0.787 | 0.825 | 0.851 | 0.872 | 0.889 | 0.899 | 0.914 | 0.039 |
| $\text{Our}_{AUDA}$ | 0.661 | 0.694 | 0.735 | 0.777 | 0.818 | 0.837 | 0.861 | 0.873 | 0.883 | 0.898 | - |
| **Ablation Studies** | | | | | | | | | | | |
| **Feature Disentanglement** | | | | | | | | | | | |
| $\text{AUDA}_{w/\mathcal{L}_1}$ | 0.615 | 0.639 | 0.687 | 0.719 | 0.781 | 0.809 | 0.819 | 0.832 | 0.843 | 0.851 | 0.01 |
| $\text{AUDA}_{w/\mathcal{L}_2}$ | 0.621 | 0.649 | 0.698 | 0.725 | 0.788 | 0.816 | 0.824 | 0.836 | 0.849 | 0.859 | 0.02 |
| $\text{AUDA}_{w/\mathcal{L}_3}$ | 0.606 | 0.637 | 0.678 | 0.707 | 0.772 | 0.796 | 0.808 | 0.819 | 0.828 | 0.841 | 0.009 |
| $\text{AUDA}_{w/\mathcal{L}_{base}}$ | 0.604 | 0.629 | 0.664 | 0.685 | 0.731 | 0.77 | 0.785 | 0.806 | 0.818 | 0.829 | 0.008 |
| **Informative Sample Selection** | | | | | | | | | | | |
| $\text{AUDA}_{w/Q_{Unc}}$ | 0.625 | 0.657 | 0.699 | 0.729 | 0.795 | 0.821 | 0.832 | 0.839 | 0.852 | 0.866 | 0.01 |
| $\text{AUDA}_{w/Q_{dom}}$ | 0.618 | 0.635 | 0.689 | 0.714 | 0.778 | 0.813 | 0.821 | 0.828 | 0.841 | 0.851 | 0.008 |
| $\text{AUDA}_{w/Q_{density}}$ | 0.610 | 0.631 | 0.685 | 0.717 | 0.77 | 0.805 | 0.812 | 0.822 | 0.831 | 0.846 | 0.009 |
| $\text{AUDA}_{w/Q_{novel}}$ | 0.600 | 0.624 | 0.682 | 0.710 | 0.767 | 0.801 | 0.809 | 0.818 | 0.829 | 0.842 | 0.004 |

Table 4: **For NIH data as the source domain**. AUC values for different baselines and proposed approach along with ablation studies. We focus on **Infiltration** condition.

## 4.4 HYPERPARAMETER SETTINGS

For our method we have two sets of hyperparameter values: for the feature disentanglement (Eqn. 1) and for informative sample selection (Eqn. 11). To set the hyperparameters for feature disentanglement we adopt the following steps using the NIH Xray dataset. For $\lambda_1$ we varied the values from $[0, 1.3]$ in steps of $0.05$, keeping $\lambda_2 = 0.45, \lambda_3 = 0.5, \lambda_{base} = 0.6$. The best classification results on a separate validation set not part of training or test datasets were obtained for $\lambda_1 = 0.85$, which was our final value. Then we vary $\lambda_2$ in a similar range with constant values of $\lambda_1 = 0.85, \lambda_3 = 0.5, \lambda_{base} = 0.6$ to get the best results for $\lambda_2 = 1.1$. We repeat the above steps to get $\lambda_3 = 0.95, \lambda_{base} = 1.2$. We repeat the entire sequence of steps for the parameters of Eqn. 11 and finally set $\lambda_{Unc} = 1.1, \lambda_{Dom} = 0.9, \lambda_{Density} = 1.05, \lambda_{Novel} = 1.25$. To address the challenge of optimizing multiple parameters we take extra precautions of multiple cross checks and using fixed seed values for reproducibility. Similar steps were followed to get parameters for CAMELYON17 and CheXpert.

## 5 DISCUSSION AND CONCLUSION

We present a novel approach for active domain adaptation that combines active learning and domain adaptation. The key motivation is to reduce the annotation cost for supervised settings, and the number of samples in unsupervised domain adaptation. We propose a novel feature disentanglement approach to obtain task specific and domain specific features. The task specific features of source and target domain are projected to a common space such that a classifier trained on one domain's features can perform equally well on the other domain. Thus active learning strategies can be used to select informative target domain samples using a classifier trained on source domain samples. We propose a novel informativeness score that selects informative samples based on the criteria of uncertainty, domainness, density and novelty. Our proposed method yields better results than SOTA methods for active learning and domain adaptation. Subsequent ablation studies also highlight the importance of each term in the loss function and justifies their inclusion. We observe that $\mathcal{L}_{base}$ is the most important comnponent of the feature disentanglement stage whereas the novelty component $Q_{novel}$ has the most contribution to selecting informative target domain samples. In future work, we aim to test our model on other medical image datasets. We also aim to test its robustness and generalizability to different classification architectures.

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
