# Active Domain Adaptation Of Medical Images Using Feature Disentanglement

## 1 Supplementary

Table 1 shows classification results for different DA techniques on CheXpert dataset as source domain and NIH dataset as target domain. The reverse scenario results are shown in the Supplementary.

| | Atel. | Card. | Eff. | Infil. | Mass | Nodule | Pneu. | Pneumot. | Consol. | Edema | Emphy. | Fibr. | PT | Hernia |
|---|---|---|---|---|---|---|---|---|---|---|---|---|---|---|
| No DA | 0.718 | 0.823 | 0.744 | 0.730 | 0.739 | 0.694 | 0.683 | 0.771 | 0.712 | 0.783 | 0.803 | 0.711 | 0.710 | 0.785 |
| MMD | 0.734 | 0.846 | 0.762 | 0.741 | 0.785 | 0.738 | 0.709 | 0.793 | 0.731 | 0.801 | 0.821 | 0.726 | 0.721 | 0.816 |
| CycleGANs | 0.751 | 0.861 | 0.785 | 0.761 | 0.817 | 0.758 | 0.726 | 0.814 | 0.746 | 0.818 | 0.837 | 0.741 | 0.737 | 0.836 |
| DANN | 0.773 | 0.882 | 0.819 | 0.785 | 0.837 | 0.791 | 0.759 | 0.838 | 0.770 | 0.836 | 0.863 | 0.766 | 0.762 | 0.861 |
| $FSL-SD$ | 0.814 | 0.929 | 0.863 | 0.821 | 0.869 | 0.825 | 0.798 | 0.863 | 0.805 | 0.872 | 0.904 | 0.802 | 0.798 | 0.892 |
| $SDA_{SOTA}$ | 0.801 | 0.913 | 0.844 | 0.807 | 0.851 | 0.809 | 0.779 | 0.848 | 0.790 | 0.849 | 0.891 | 0.789 | 0.781 | 0.873 |
| $UDA_{SOTA}$ | 0.786 | 0.906 | 0.833 | 0.789 | 0.831 | 0.802 | 0.763 | 0.835 | 0.774 | 0.837 | 0.868 | 0.768 | 0.763 | 0.860 |

Table 1: Classification results on the NIH Xray dataset's test split using CheXpert data as the source domain. Note: $FSL-SD$ is a fully-supervised model trained on target domain data.

### 1.1 Robustness and Generalization

To test the robustness of the proposed approach, we added simulated noise of $\mu = 0$ and different $\sigma \in \{0.005, 0.01, 0.015, 0.05, 0.1\}$ and run our UDA pipeline. Figure 1 shows the AUC values for the baseline performance of UDA and different $\sigma$. The results are close to UDA for $\sigma = 0.005, 0.01$, but start to degrade significantly for noise levels above $\sigma = 0.01$, which we term as noise threshold. These results demonstrate that our method is robust to a reasonable level of added noise.

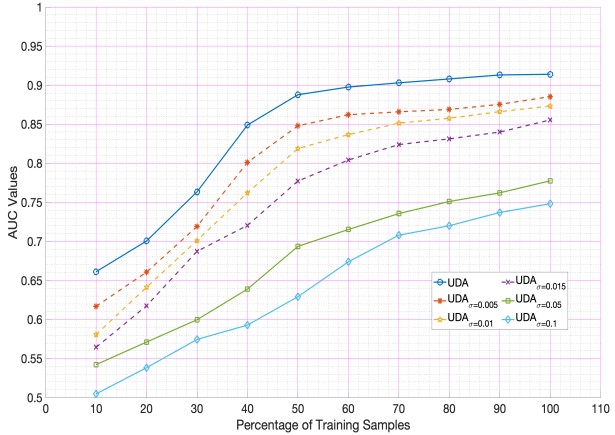

Figure 1: AUC measures for different features for added Gaussian noise of $\mu = 0$ and different $\sigma$.

### 1.2 T-SNE Visualizations

Figure 2 (a) shows the t-sne plots of image features (taken from the fully connected layer of a DenseNet-121 trained for image classification) while Figure 2 (b) shows the plot using the class-specific features. The plots of the original features shows different image class clusters that overlap and that makes it challenging to have good classification. On the other hand, the clusters obtained

| | 10% | 20% | 30% | 40% | 50% | 60% | 70% | 80% | 90% | 100% | p- |
|---|---|---|---|---|---|---|---|---|---|---|---|
| FSL-SD | 0.821 | 0.821 | 0.821 | 0.821 | 0.821 | 0.821 | 0.821 | 0.821 | 0.821 | 0.821 | < 0.001 |
| Random | 0.632 | 0.663 | 0.702 | 0.736 | 0.768 | 0.775 | 0.792 | 0.803 | 0.811 | 0.814 | < 0.001 |
| Unc | 0.641 | 0.678 | 0.719 | 0.757 | 0.789 | 0.801 | 0.812 | 0.825 | 0.836 | 0.843 | < 0.001 |
| AADA | 0.649 | 0.686 | 0.728 | 0.768 | 0.80 | 0.821 | 0.837 | 0.851 | 0.867 | 0.873 | < 0.001 |
| BADGE | 0.638 | 0.672 | 0.714 | 0.751 | 0.785 | 0.804 | 0.817 | 0.834 | 0.843 | 0.851 | 0.005 |
| CLUE | 0.641 | 0.68 | 0.721 | 0.761 | 0.789 | 0.812 | 0.830 | 0.843 | 0.859 | 0.862 | 0.007 |
| Fu | 0.649 | 0.686 | 0.728 | 0.768 | 0.80 | 0.821 | 0.837 | 0.851 | 0.867 | 0.873 | 0.001 |
| Su | 0.651 | 0.683 | 0.725 | 0.773 | 0.802 | 0.818 | 0.835 | 0.857 | 0.866 | 0.877 | 0.02 |
| Our$_{ASDA}$ | 0.661 | 0.696 | 0.737 | 0.78 | 0.817 | 0.843 | 0.865 | 0.881 | 0.891 | 0.907 | 0.039 |
| Our$_{AUDA}$ | 0.657 | 0.689 | 0.730 | 0.772 | 0.811 | 0.829 | 0.855 | 0.869 | 0.878 | 0.892 | - |
| **Ablation Studies** | | | | | | | | | | | |
| **Feature Disentanglement** | | | | | | | | | | | |
| AUDA$_{w/\mathcal{L}_1}$ | 0.611 | 0.634 | 0.681 | 0.714 | 0.775 | 0.803 | 0.814 | 0.828 | 0.838 | 0.847 | 0.01 |
| AUDA$_{w/\mathcal{L}_2}$ | 0.618 | 0.645 | 0.692 | 0.721 | 0.784 | 0.811 | 0.82 | 0.831 | 0.844 | 0.853 | 0.02 |
| AUDA$_{w/\mathcal{L}_3}$ | 0.603 | 0.632 | 0.673 | 0.702 | 0.767 | 0.791 | 0.804 | 0.814 | 0.822 | 0.835 | 0.009 |
| AUDA$_{w/\mathcal{L}_{base}}$ | 0.601 | 0.628 | 0.662 | 0.683 | 0.735 | 0.772 | 0.789 | 0.801 | 0.813 | 0.826 | 0.008 |
| **Informative Sample Selection** | | | | | | | | | | | |
| AUDA$_{w/Q_{Unc}}$ | 0.621 | 0.654 | 0.695 | 0.725 | 0.792 | 0.819 | 0.826 | 0.836 | 0.849 | 0.861 | 0.01 |
| AUDA$_{w/Q_{dom}}$ | 0.612 | 0.633 | 0.685 | 0.711 | 0.772 | 0.809 | 0.816 | 0.825 | 0.837 | 0.849 | 0.008 |
| AUDA$_{w/Q_{density}}$ | 0.605 | 0.628 | 0.681 | 0.712 | 0.767 | 0.801 | 0.809 | 0.818 | 0.828 | 0.841 | 0.009 |
| AUDA$_{w/Q_{novel}}$ | 0.595 | 0.621 | 0.677 | 0.706 | 0.762 | 0.795 | 0.801 | 0.813 | 0.824 | 0.836 | 0.004 |

Table 2: **For CheXpert data as the source domain**. AUC values for different baselines and proposed approach along with ablation studies. We focus on **Infiltration** condition.

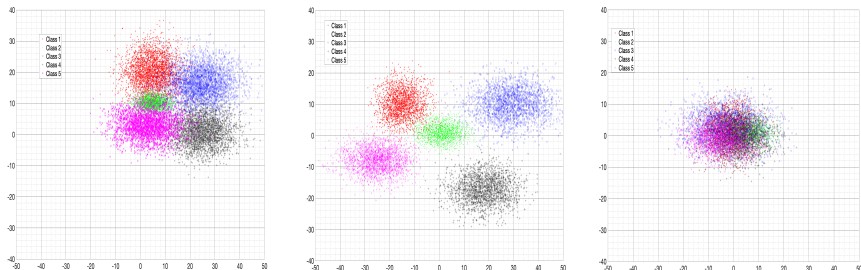

Figure 2: T-sne results comparison between original image features and feature disentanglement output of source domain features. (a) Original image features; (b) Task specific features; (c) Domain specific features.

using the task-specific features are well separated and there is less overlap between different clusters. Figure 2 (c) shows the output of using domain specific features where a significant overlap is observed among classes. This clearly demonstrates the efficacy of our feature disentanglement method, i.e., the task-specific and domain specific features fulfil their desired objectives. In the example in Figure 2, the features are taken from images belonging to 5 classes (Atelectasis,Consolidation, Effusion, Infiltration and Nodule) from the NIH dataset.

## 1.3 BRAIN AGE ESTIMATION

We use our method for brain age prediction across different domains. We take the work by More et al. (2023) as our baseline, and we apply our method to the different dataset combinations of CanCAM Taylor et al. (2017),IXI[1], eNKI Nooner & et. al (2012) and 1000Brains Caspers & et. al (2014) and report the results in Table 3. We closely follow the settings described in More et al. (2023) and report results for different cross-dataset settings. In this setting 3 datasets are used for training and the 4th one for testing. For our we use three datasets separately as source domain while keeping the 4th dataset constant as target domain. For a given target domain dataset (e.g. CanCAM) the final numbers are an average of 3 runs where each of th other three dataset (e.g., IXI, eNKI and 1000Brains) was the source dataset. This is repeated for all 4 datasets being the **target domain**.

---

[1]http://brain-development.org/ixi-dataset/

| Target Domain | MAE | | MSE | | $R^2$ | | Corr(true, pred) | | Threshold % |
|---|---|---|---|---|---|---|---|---|---|
| Dataset | Our | Baseline | Our | Baseline | Our | Baseline | Our | Baseline | |
| CamCAN | 4.49 | 4.75 | 36.02 | 38.35 | 0.91 | 0.89 | 0.96 | 0.95 | 34% |
| IXI | 5.69 | 6.08 | 50.2 | 57.35 | 0.82 | 0.79 | 0.94 | 0.94 | 49% |
| eNKI | 4.71 | 4.97 | 34.87 | 39.65 | 0.92 | 0.88 | 0.94 | 0.94 | 41% |
| 1000-BRAINS | 4.84 | 5.13 | 38.72 | 41.03 | 0.79 | 0.73 | 0.91 | 0.90 | 57% |

Table 3: **Brain Age Prediction Results** . Abbreviations: $MAE$: mean absolute error between true and predicted age, $MSE$: mean squared error between true and predicted age, $R^2$ : the proportion of variance of predicted age explained by the independent variables in the model, $Corr(true, pred)$: Pearson's correlation between true and predicted age.

We observe that our method gives better performance than the numbers reported in More et al. (2023) with lower values of MAE and MSE, and higher values of $R^2$. We also report a value under "Threshold %" which is the percent of labeled target domain samples that were required to beat the baseline numbers. It clearly shows that our proposed method gets better results with fewer labeled samples and our approach does a good job in identifying informative samples in the presence of domain shifts.

## 1.4 ACTIVE DOMAIN ADAPTATION FOR SEGMENTATION

We also apply our method for segmentation. For segmentation purposes, inspired by Park et al. (2020), we change our feature disentanglement network such that the task specific component is a spatial map of size $64 \times 64$, which is then resized to the original image size. The segmentation network is a UNet Ronneberger et al. (2015) which is trained on this spatial feature maps of source domain images instead of the original images. Following the steps in Nath et al. (2021) for active learning based segmentation we select informative samples from the target domain using the task specific spatial map and use it for source free domain adaptation using the method described in Bateson et al. (2022). The method is applied on the MICCAI 2018 IVDM3Seg Challenge dataset. The baseline method of Bateson et al. (2022) gets a dice similarity score (DSC) of $74.2$, whereas we obtain a DSC of 76.3 using $56\%$ of the labeled data. This clearly indicates that using our method we obtain better results with fewer labeled samples, despite the domain shift.