# OpenReview forum: "Active Domain Adaptation Of Medical Images Using Feature Disentanglement"
_ICLR.cc/2024/Conference — Submitted to ICLR 2024_

### Official Review · Reviewer_Hvpz · 2023-10-19

**Soundness:** 1 poor
**Presentation:** 3 good
**Contribution:** 1 poor
**Rating:** 3
**Confidence:** 4

**Summary:**

The authors propose a method for combining active learning techniques with domain adaptation. They propose an algorithm for learning task-specific and task-shared features, along with several metrics which are supposed to quantify informativeness of samples for active learning. They evaluate their method on 2 datasets from the medical imaging domain, showing an improvement over the baselines.

**Strengths:**

- Authors performed ablation studies for all proposed modifications
- The proposed approach achieves (slightly) better results than baselines

**Weaknesses:**

1. The paper seems written in a rush and it’s difficult to read at times
2. Using features of a pre trained classifier for L_base does not need to identify the correct task-specific features. Nothing prohibits the model from a) extracting task-independent features and assigning them zero weights in the final classification layer and b) collapsing to the target variable already in the hidden layer (or the logit distribution)
3. The density estimation approach is incorrect. You cannot reason about probability densities by comparing cosine similarities of **arbitrarily distributed** vectors (e.g., imagine the case where several dimensions are strongly correlated).
4. In fact most of your objectives suffer from that same problem - the similarities can easily be inflated if multiple dimensions are not independent. Consider using a more probabilistically sound approach, e.g., by incorporating models such as normalizing flows

**Questions:**

1. “Given source and target domains S and T, an ideal domain independent feature’s classification accuracy on domain S is close to those obtained using the original images’ features.” - I do not understand this sentence
2. How did you select the hyperparameters (e.g., the percentiles for similarity cutoffs)? Actually looks like you just fitted the hyperparameters to the final results? Which is incorrect? How did you select hyperparameters for the baselines ?

---

> ### Author Response · Authors · 2023-11-18
> **Rebuttal to Official Review of Submission7450 by Reviewer Hvpz**
>
> Weaknesses 1: The paper seems written in a rush and it’s difficult to read at times
>
> Our Response: We regret that our writing was not clear to the Reviewer.We will proofread the manuscript again and make sure to improve the readability of the final version.
>
> Weakness 2: Using features of a pre trained classifier for L_base does not need to identify the correct task-specific features. Nothing prohibits the model from a) extracting task-independent features and assigning them zero weights in the final classification layer and b) collapsing to the target variable already in the hidden layer (or the logit distribution)
>
> Our Response: We disagree with the reviewer.It is known in domain generalization that when finetuning models on a small amount of data from a different domain the output features tend to collapse in diversity.See section 2 of: Wuyang Chen et. al, "CONTRASTIVE SYN-TO-REAL GENERALIZATION".Using L_base helps us prevent such feature collapse.While using M_base we assume that the classifier is the optimal one. Thus the resulting set of features from M_base’s penultimate layer (not the logit distribution) are expected to be the optimal features.
>
> While reviewer's point (b) can possibly occur, the use of dropout after the feature layer mitigates such issues. L_base is not the only criterion for getting task specific and domain specific features. Combined with the other loss functions L_1,L_2,L_3, we ensure that the final z_task is not prone to inaccuracies.
>
> Weakness 3: The density estimation approach is incorrect. You cannot reason about probability densities by comparing cosine similarities of arbitrarily distributed vectors (e.g., imagine the case where several dimensions are strongly correlated).
>
> Our Response: We believe there is a misunderstanding by the reviewer. We are not using density estimation/probability values in any of our proposed loss functions. The only instance is use of predictive entropy for Q_Unc in Eqn. 7, which is a standard technique.
> Can the reviewer be more specific to which equation they are referring to?
>
> Weakness 4 : In fact most of your objectives suffer from that same problem - the similarities can easily be inflated if multiple dimensions are not independent. Consider using a more probabilistically sound approach, e.g., by incorporating models such as normalizing flows
>
> Our Response: We like to clarify that  we use an autoencoder, as described in the text and different loss functions- e.g. reconstruction loss is used for autoencoder training. An autoencoder is a better option than normalizing flows for feature disentanglement since we do not need density distributions.  (A note is that in Fig 1a of the paper I show VAE encoder. In reality it is simple autoencoder since I use only th reconsrucion loss. Perhaps the reviewer is confused because of that?)
>
> Questions 1: “Given source and target domains S and T, an ideal domain independent feature’s classification accuracy on domain S is close to those obtained using the original images’ features.” - I do not understand this sentence
>
> Our Response: Let us consider an optimal classification network (M_optimal) which gives the best results for a particular dataset which is part of the source domain S. Our motivation is to obtain domain independent features that give a similar performance on this source domain dataset.  We can rephrase the sentence as “Given source and target domains S and T, an ideal domain independent feature's classification accuracy on domain S is close to those obtained using features from an optimal classifier.”
>
> Question 2: How did you select the hyperparameters (e.g., the percentiles for similarity cutoffs)? Actually looks like you just fitted the hyperparameters to the final results? Which is incorrect? How did you select hyperparameters for the baselines ?
>
> Our response: The percentile based values for eta_1 and eta_2 were chosen based on steps similar to what one would choose for other parameters. Insead of having a fixed value for these parameters we observe that a good threshold is highly dependent on the actual similarity values. Hence we experimented with different percentile values as threshold varying them from 5% to 60% for eta_1, and 20% to 90% for eta_2, across all datasets. We found that eta_1=30% and eta_2=75% gives the best results across all datasets.
>
> We have also selected the hyperparameters after careful consideration. We confirm that as per standard practices in machine learning the parameter values were set using a separate validation set which was not part of training or test datasets. We agree that this is a high number of parameters that need to be optimized due to the multiple loss terms in our method. Hence we have taken extra care to ensure that the optimization steps were done with utmost care and we have cross checked them multiple times. Reproducible results were ensured with fixed seed values. The high number of hyperparameters leads to much improved results.

---

> > ### Comment · Reviewer_Hvpz · 2023-11-20
> >
> > Weakness 2:
> >
> > Section2 of the linked paper just shows a “motivating example” on a toy dataset, it is by no means a proof that NNs converge to diverse features in the penultimate layer. Even when using dropout the model can in principle just learn to duplicate a feature N times.
> >
> > Weakness 3 and 4:
> >
> > You are not using probabilities, yes, but you reason about similarities of vectors with your proposed approach. I do not see how your approach can actually guarantee you will select truly similar samples.

---

> ### Author Response · Authors · 2023-11-23
> **Response to Official Comment by Reviewer Hvpz**
>
> Thank you for the comments.
> 1. The strategy of turning off connections in dropout makes it less likely that the learned features will be duplicates of other features.
>
> 2. Although no method can guarantee the selection of highly similar samples, the cosine similarity coupled with our multiple constraints can be expected to do better than state of the art methods. This hypothesis is supported by our method's performance.

---

> > ### Comment · Reviewer_Hvpz · 2023-11-23
> >
> > 1. I do not see how this is true, at least without a proper proof. I believe the opposite is true, since features can be turned off at random, the important ones will have to be duplicated in order to retain performance.
> >
> > 2. Again, I do not think this is true. If you were to use a probabilistic approach for example you would have theoretical guarantees.
> > Your improved empirical performance (in terms of predictions) doesn’t necessarily guarantee you are selecting actually similar  samples. This might still happen of course, but is not guaranteed, as you did not provide any theoretical justification for it

---

> > > ### Author Response · Authors · 2023-11-23
> > > **Response to Official Comment by Reviewer Hvpz**
> > >
> > > Thank you for your feedback
> > >
> > > 1. If dropout were to lead to duplication of features then it would defeat its primary purpose of preventing overfitting and improving generalization power of the network.
> > >
> > > 2. For our purpose of feature disentanglement it is better to work with feature vectors than probability distributions since feature vectors are easily interpretable and capture the essential features of the data.

---

### Official Review · Reviewer_FaUD · 2023-10-31

**Soundness:** 3 good
**Presentation:** 3 good
**Contribution:** 3 good
**Rating:** 6
**Confidence:** 4

**Summary:**

This work proposes a novel strategy for performing domain adaptation in an active learning scenario.
The method is based on learning disentangled representations referred to domain and task, from which an informative score is computed on samples from the target domain. The most informative samples (below a certain available budget) are chosen and added to the training set.

**Strengths:**

- The proposed method is interesting and seems to provide good performance

- The problem tackled is very relevant in the medical field due to the cost of annotating data

**Weaknesses:**

- The proposed method is not simple in terms of optimizations: 5 different losses are used, each with its own hyperparemeter. Also the informative score introduces many hyperparamers. Choosing many hyperparameters is not trivial, and authors report some arbitrary value for them. How were they chosen? It is not completely clear

- The tables are hard to read, best results are not highlighted. I also suggest author report a standard devation, especially in the higher p-value cases

- I think the experimental validation is somewhat lacking, as only two settings were explored (histology and cxr images). I suggest authors also include other modalities or tasks. For example brain MRI with the task of brain age regression (particularly relevant for domain shift), or image segmentation.

**Questions:**

See weaknesses.

Additional questions:

- Could your framework be adapted to other tasks such as segmentation or regression (as mentioned in the weaknesses)?

- In your experimental protocol (Sec. 4.1) you select at each step 10% of the size of the training data from the target set. This means that the added samples will be in minority in the training set. Have you also tried reweighting them in the classification loss? Can this help in reducing the number of samples required for better AUC?

- For improving the readability of this work, I think that the results could be presented in from of plots of AUC vs size rather then big tables

---

> ### Author Response · Authors · 2023-11-18
> **Rebuttal to Official Review of Submission7450 by Reviewer FaUD**
>
> Weaknesses 1:
> The proposed method is not simple in terms of optimizations: 5 different losses are used, each with its own hyperparemeter. Also the informative score introduces many hyperparamers. Choosing many hyperparameters is not trivial, and authors report some arbitrary value for them. How were they chosen? It is not completely clear
>
> Our Response: We would like to clarify that we select the hyperparameters after careful consideration. We confirm that as per standard practices in machine learning the parameter values were set using a separate validation set which was not part of training or test datasets. We agree that this is a high number of parameters that need to be optimized due to the multiple loss terms in our method. Hence we have taken extra care to ensure that the optimization steps were done with utmost care and we have cross checked them multiple times. Reproducible results were ensured with fixed seed values. The high number of hyperparameters leads to much improved results.
>
> Weakness 2:  The tables are hard to read, best results are not highlighted. I also suggest author report a standard deviation, especially in the higher p-value cases
>
> Our Response: Due to space constraints we did not report standard deviations. However we will include them in the final manuscript. The standard deviations among close performing methods is such that there is negligible overlap in their values, especially methods with higher p-values
>
> Weakness 3: I think the experimental validation is somewhat lacking, as only two settings were explored (histology and cxr images). I suggest authors also include other modalities or tasks. For example brain MRI with the task of brain age regression (particularly relevant for domain shift), or image segmentation.
>
> Our Response: We believe that we have done exhaustive validation on 3 different datasets for medical image classification, which has been acknowledged by other reviewers. Our method may work for brain age regression without major changes, but will need major changes for segmentation. Since domain adaptive segmentation  would require the training step to learn structural information, we need to change our method.
>
> Additional Question 2: In your experimental protocol (Sec. 4.1) you select at each step 10% of the size of the training data from the target set. This means that the added samples will be in minority in the training set. Have you also tried reweighting them in the classification loss? Can this help in reducing the number of samples required for better AUC?
>
> Our Response: We believe the reviewer is alluding to importance sampling. While updating the classifier we use the labels of the newly added samples to finetune the classifier. There is no explicit reweighting, although that can be explored in future work.
>
> Additional Question 2: For improving the readability of this work, I think that the results could be presented in from of plots of AUC vs size rather then big tables
>
> Our Response: We thank the Reviewer for their suggestion and agree this change might be a good way to represent our results.  We will add the figures to the final version’s supplementary material.

---

> > ### Comment · Reviewer_FaUD · 2023-11-21
> >
> > I thank the authors for their response. However I still think that more empirical validation is needed on different medical tasks, hence I confirm my current rating

---

> > > ### Author Response · Authors · 2023-11-22
> > > **Response to second comments by Reviewer FaUD**
> > >
> > > We thank the reviewer for their suggestions. We have been conducting experiments on brain age prediction and medical image segmentation, and the updated results are included in the updated Supplementary document. We hope that the extra results can convince the reviewer about the applicability of our method in different medical image analysis settings.

---

> ### Comment · Reviewer_FaUD · 2023-11-22
>
> I thank the authors for the additional experiments and results. I have read the supplementary material and I think the experimental validation is now more convincing. The new experiments should be mentioned in the revised main text. I gladly raise my score.

---

### Official Review · Reviewer_nM77 · 2023-11-01

**Soundness:** 4 excellent
**Presentation:** 3 good
**Contribution:** 3 good
**Rating:** 6
**Confidence:** 4

**Summary:**

This paper presents an innovative active learning method for domain adaptation. The problem at hand involves two data domains: the source and target domains, with a distribution shift between them. The algorithm comprises two key steps. In the first step, data (images) in both domains are transformed into a latent space using two separate autoencoders, and the feature representations in the latent space are disentangled into domain-specific and task-specific representations. It is assumed that the domain-specific representations account for the distribution shift. In the second step, criteria were designed to select informative unlabeled image samples in the target domain for labeling. The labeled image samples are then added to the labeled image samples from the source domain to update the classification model trained on the labeled data from the source domain. This process can be repeated multiple times. The proposed method was evaluated on two public image analysis datasets and outperformed several state-of-the-art active learning methods and a couple of domain adaptation algorithms.

**Strengths:**

1. Incorporating active learning, feature disentanglement, and domain adaptation all together seems to be an innovative idea.
2. The proposed metrics for identifying informative image samples based on disentangled feature representations appear to be effective.
3. The proposed algorithm was evaluated on two relatively large medical imaging datasets, and it achieved superb results.

**Weaknesses:**

1. The two medical image datasets are quite large. In the experimental setting, at least 10% of the unlabeled data samples in the target domain were selected for labeling and added to the source domain's labeled data to update the classifier. While 10% still represents a significant number, considering the constraints in the research setting—such as limited and expensive expertise in the medical field—it would be more valuable to evaluate the effectiveness of the proposed method with much fewer labeled samples from the target domain, for example, 0.1% or 1%. Unfortunately, this aspect is missing in the paper.

2. The discussion section or the presentation of the paper could be improved. For instance, in the ablation study, each component of the loss function and the metrics for identifying informative data samples was thoroughly examined, and their contributions were reported in the tables. However, there is a lack of in-depth discussion and no clear claims have been made regarding which component contributes more. It is not readily apparent to readers which component has the most significant impact.

**Questions:**

In the algorithm description, the authors initially state that feature disentangling was performed jointly using data samples from both the source and target domains. However, they later mention that the process was performed using data solely from the source domain. Which statement is correct?

---

> ### Author Response · Authors · 2023-11-18
> **Rebuttal to Official Review of Submission7450 by Reviewer nM7**
>
> Weaknesses 1:
> The two medical image datasets are quite large. In the experimental setting, at least 10% of the unlabeled data samples in the target domain were selected for labeling and added to the source domain's labeled data to update the classifier. While 10% still represents a significant number, considering the constraints in the research setting—such as limited and expensive expertise in the medical field—it would be more valuable to evaluate the effectiveness of the proposed method with much fewer labeled samples from the target domain, for example, 0.1% or 1%. Unfortunately, this aspect is missing in the paper.
>
> Our Response: We thank the Reviewer for raising this point. Although we show results for every 10% increment to the training data, it is done over batches. We use a batch size of 96 in determining the informative samples. On an average in every batch 2-3 samples  of each class are chosen  to add to the training set. This represents 0.1% or less for each class in the NIH dataset. The increase of performance is very low over batches and becomes prominent after every 10% increment of dataset size. In the final manuscript we will provide results for 0.1% and 1% labeled target samples.
>
> Weakneess 2: The discussion section or the presentation of the paper could be improved. For instance, in the ablation study, each component of the loss function and the metrics for identifying informative data samples was thoroughly examined, and their contributions were reported in the tables. However, there is a lack of in-depth discussion and no clear claims have been made regarding which component contributes more. It is not readily apparent to readers which component has the most significant impact.
>
> Our Response: We agree that our discussions could be improved and we should emphasize the most important components of our method. We have addressed this issue in the revised manuscript and  will elaborate further in the final manuscript.
>
> Question 1: In the algorithm description, the authors initially state that feature disentangling was performed jointly using data samples from both the source and target domains. However, they later mention that the process was performed using data solely from the source domain. Which statement is correct?
>
> Our Response: We apologize for the confusion. The feature disentanglement network was trained on source and target data. The confusion possibly arises from the text in Sec 4.1. We would like to clarify that in Sec 4.1 we refer to those target domain  samples that were not part of the training process.

---

> > ### Comment · Reviewer_nM77 · 2023-11-22
> > **Thank you for your response.**
> >
> > Thank you for your response.
> >
> > Reviewer: nM77

---

### Official Review · Reviewer_S5xV · 2023-11-01

**Soundness:** 3 good
**Presentation:** 3 good
**Contribution:** 3 good
**Rating:** 6
**Confidence:** 4

**Summary:**

This paper presents a method that uses feature disentanglement for active learning. The method is demonstrated under domain shift, i.e. actively selecting examples for further training so as to adapt to a shifted target domain. Results are given on multi-centre histopathology and chest x-ray datasets.

**Strengths:**

The method involves an informativeness score that combines measures of uncertainty, “domainness”, density, and novelty. There is some novelty in this.
Fairly extensive experiments are reported incorporating 6 methods from the literature on two medical applications using public datasets. Overall the performance seems promising.
Under “Ablation Studies” each of the free parameters in the loss and informativeness score (Eqns (10 and (11)) is set to zero in turn and the effect on performance measured. This is a useful experiment to show that each term has an effect (although in a few cases removal of L_1 or Q_unc seems to have helped, and that could be commented upon).

**Weaknesses:**

My main criticism is that the method has 4 free parameters in the loss function (Equation (1)) and another 4 in the informativeness score (Equation (11)). This is a high number of hyperparameters to set empirically and it needs to be clear that this has been done carefully and reproducibly. For the histopathology and CheXpert experiments, values are stated without any explanation of how these values were arrived at. This needs some comment, and in particular we need to know for certain that these values were determined without using test data in any way. For the NIH ChestXray experiment, subsection 4.4 describes a greedy hyperparameter search; again it needs to be clarified that test data were not used in this search (presumably). If test performance was used in this search then the results would be invalid. Hopefully this is not the case.

**Questions:**

See above.

---

> ### Author Response · Authors · 2023-11-18
> **Rebuttal to Official Review of Submission7450 by Reviewer S5xV**
>
> We thank the Reviewer for their comments. Indeed we can confirm that as per standard practices in machine learning the parameter values were set using a separate validation set which was not part of training or test datasets.
>
> We agree that this is a high number of parameters that need to be optimized due to the multiple loss terms in our method. Hence we have taken extra care to ensure that the optimization steps were done with utmost care and we have cross checked them multiple time. Reproducible results were ensured with fixed seed values.
>
> To get the values for Histopathology dataset (CAMELYON17) and CheXpert, we follow the same procedure as the NIH dataset. We have mentioned these points in the revised manuscript.

---

> > ### Comment · Reviewer_S5xV · 2023-12-04
> >
> > Thankyou for your response and for planning to address my comment about hyperparameter search by making the procedure clear in the revision for all three datasets. My rating remains the same.

---

### Meta-Review · Area_Chair_YKK4 · 2023-12-05

**Metareview:**

This paper received contrasting reviews, scoring 6, 6, 6, and 3. These are the final evaluations after rebuttal and discussions.

The paper is recognized to be sufficiently well structured and presenting an effective method, with good performances.
Main criticisms regard on how to set the free hyper-parameters for the datasets considered, and some issues concerning the results, especially ablations, such as insufficient analysis of the loss function to figure out which is the term with the most effective contribution, the percentage of validation data used, missing standard deviations when reporting the numerical performances.
Other remarks concern the low clarity of the writing in general and unclear presentation of some stages of the method, which prevented the full understanding of the work, including the lack of a theoretical analysis.
The rebuttal addressed all these aspects, not all in a well detailed manner, and a reviewer remained unsatisfied while the others just fairly acknowledged the given answers, being mildly positive.

The AC read the comments, solicited a discussion with reviewers, and discussed carefully this paper with SAC. Both AC and SAC went through the paper to better understand pros and cons of the work, and after a deep discussion, they agreed that the paper in this form is not yet ready for acceptance to ICLR 2024.

The reasons are mainly due to the following aspects. While the approach presents some original aspects validated by good performance, it is mainly procedural with weak justifications of all the designed steps. The lack of theoretical justifications, which is an issue for a reviewer, is not particularly felt as such by the AC and SAC, but when also the empirical analysis is lacking in some respect, there is an issue.
In fact, what is considered a problem is instead the lack of a convincing ablation analysis, especially the setting of the hyper-parameters, for which it is not sufficient to only assert that they have been found by an independent validation set. The overall method and the experimental protocol are scattered of parameters and hyper-parameters and performances seem really quite dependent on their settings.

**Justification For Why Not Higher Score:**

Reviews are not unanimous.
Discussion between AC and SAC led to rejection

**Justification For Why Not Lower Score:**

N/A

---

### Decision · Program_Chairs · 2024-01-16

Reject